# The SWI/SNF chromatin remodelling complex is required for maintenance of lineage specific enhancers

Burak H. Alver[1,*], Kimberly H. Kim[2,3,4,*,†], Ping Lu[2,3,4,*], Xiaofeng Wang[2,3,4], Haley E. Manchester[2,3,4], Weishan Wang[2,3,4], Jeffrey R. Haswell[2,3,4], Peter J. Park[1] & Charles W.M. Roberts[2,3,4,5]

Genes encoding subunits of SWI/SNF (BAF) chromatin remodelling complexes are collectively altered in over 20% of human malignancies, but the mechanisms by which these complexes alter chromatin to modulate transcription and cell fate are poorly understood. Utilizing mouse embryonic fibroblast and cancer cell line models, here we show via ChIP-seq and biochemical assays that SWI/SNF complexes are preferentially targeted to distal lineage specific enhancers and interact with p300 to modulate histone H3 lysine 27 acetylation. We identify a greater requirement for SWI/SNF at typical enhancers than at most super-enhancers and at enhancers in untranscribed regions than in transcribed regions. Our data further demonstrate that SWI/SNF-dependent distal enhancers are essential for controlling expression of genes linked to developmental processes. Our findings thus establish SWI/SNF complexes as regulators of the enhancer landscape and provide insight into the roles of SWI/SNF in cellular fate control.

[1] Department of Biomedical Informatics, Harvard Medical School, Boston, Massachusetts 02115, USA. [2] Department of Pediatric Oncology, Dana-Farber Cancer Institute, Boston, Massachusetts 02215, USA. [3] Division of Hematology/Oncology, Children's Hospital, Boston, Massachusetts 02215, USA. [4] Department of Pediatrics, Harvard Medical School, Boston, Massachusetts 02215, USA. [5] Comprehensive Cancer Center and Department of Oncology, St. Jude Children's Research Hospital, Memphis, Tennessee 38105, USA. * These authors contributed equally to this work. † Present address: Department of Oncology Target Discovery, Pfizer, Pearl River, New York 10965, USA. Correspondence and requests for materials should be addressed to P.J.P. (email: peter_park@hms.harvard.edu) or to C.W.M.R. (email: charles.roberts@stjude.org).

SWI/SNF (BAF) chromatin remodellers are large ($\sim$2 MDa), evolutionarily conserved complexes, each composed of approximately 15 protein subunits[1]. Of these subunits, several core members are present in most or all SWI/SNF complexes, including SMARCB1 (also known as SNF5, BAF47 and INI1), SMARCC1/SMARCC2 (also known as BAF155 and BAF170), and one of the two mutually exclusive ATPase subunits, SMARCA4 (also known as BRG1) and SMARCA2 (also known as BRM), which utilize energy derived from ATP hydrolysis to mobilize nucleosomes[2,3]. In addition, SWI/SNF complexes typically include a number of lineage-restricted subunits that vary by cell type[4–7], and are likely crucial to the specific function of these complexes. With mutations in genes encoding SWI/SNF subunits (for example, *ARID1A*, *ARID1B*, *ARID2*, *PBRM1*, *SMARCA4* and *SMARCB1*) collectively occurring in $\sim$20% of all tumours whose genomes have been characterized to date, SWI/SNF complexes are the most commonly mutated chromatin modulators in human cancer[8].

Although SWI/SNF complexes have been found to serve key roles in transcriptional regulation[9] and tumour suppression, the mechanism by which SWI/SNF complexes execute these functions remains poorly understood. Previous work has shown that SWI/SNF plays a role in chromatin remodelling at both promoters[10] and enhancers[11], and is required for the activity of certain enhancers that are important for cell identity[12–17]. We have also previously found that oncogenic transformation following the loss of the known tumour suppressor subunit SMARCB1 involves an imbalance in an antagonistic relationship between mammalian SWI/SNF complexes and polycomb repressive complex 2 (PRC2)[18]. PRC2 catalyses histone 3 lysine 27 trimethylation (H3K27me3) via the methyltransferase catalytic subunit, EZH2, and thereby promoting transcriptional silencing.

Enhancers are DNA sequences consisting of binding sites of transcription factors (TFs) that facilitate the activation of target genes. In addition to various cell type-specific TFs such as master regulators, enhancers are also bound by general factors such as the mediator complex, cohesion complex and various coactivators[19,20]. Among these coactivators are enzymes that remodel nucleosomes or post-translationally modify histones to sustain an accessible chromatin state. In particular, enhancers are highly enriched with H3K4me1, and H3K4me2, in contrast to active promoters that are enriched with H3K4me3 (refs 21,22). Both enhancers and promoters are enriched in acetylation marks such as H3K27ac when active[23–25]. Importantly, chromatin signatures at enhancers tend to be much more variable across different cell types compared with promoters, implicating them in the modulation of cell type-specific gene expression levels[26]. A sub-class of enhancers that consist of clusters of highly active enhancers, defined as super-enhancers, have been implicated in the control of master regulators of cell identity[20].

In the work presented here, we set out to build on our previous work and evaluate the role of SWI/SNF in control of methylation of H3K27 by EZH2. This leads to our observation that the SWI/SNF complex plays a major role in regulating H3K27 acetylation (H3K27ac) in both loss-of-function mouse embryonic fibroblast (MEF) and gain-of-function rhabdoid tumour (RT) systems. By characterizing the *cis*-regulatory elements that are affected by the loss of different SWI/SNF subunits, we observe that enhancers have a greater requirement for the SWI/SNF complex than promoters. Surprisingly, we find that typical enhancers are more strongly affected than super-enhancers. Mechanistically, we show that the regulation of H3K27ac at enhancers by SWI/SNF happens via its interaction with the p300/CBP histone acetyl-tranfrase. Furthermore, we show that enhancers regulated by the SWI/SNF complex are essential in controlling genes important for lineage specification.

## Results

**The role of SWI/SNF for H3K27 acetylation of enhancers.** We utilized a mouse embryonic fibroblast (MEF) system in which the *Smarca4*, *Smarcb1* or *Arid1a* SWI/SNF subunit genes could be conditionally deleted, as well as *Ezh2* conditional knockout MEFs to gauge PRC2 function (Methods). Although we noted a subtle increase in global H3K27me3 levels upon SWI/SNF subunit inactivation in western blots, surprisingly a much greater effect of decreased global levels of H3K27ac was observed (Fig. 1a and Supplementary Fig. 1). In contrast to the marked effects upon H3K27 acetylation, SWI/SNF subunit loss minimally affected the total H3 acetylation, total H4 acetylation, and total histone levels (Supplementary Fig. 1).

To localize the changes in H3K27ac and to evaluate them in the context of DNA regulatory elements, we performed chromatin immunoprecipitation followed by sequencing (ChIP-seq) for H3K27ac in wild-type MEFs and in *Smarca4* or *Smarcb1* conditional knockout MEFs. We identified regions enriched for H3K27ac and classified them based on their proximity to active transcription start sites (TSS) marked by H3K4me3. A total of 11,531 sites proximal to active TSSs and 21,772 sites distal to active TSSs were identified as putative promoters and enhancers, respectively. We validated that the enhancer set defined in this work is consistent with enhancers identified in MEFs by the Mouse Encyclopedia of DNA Elements (ENCODE) consortium[27] (Methods, Supplementary Fig 2). Following inactivation of either *Smarcb1* or *Smarca4*, we observed little change in H3K27ac levels at promoters, but a marked reduction at many enhancers (Fig. 1b,c). This reduction at enhancers was robust and consistent across replicate experiments performed on MEFs derived from independent mice (Methods, Supplementary Figs 3-7). Furthermore, the signal changes at enhancers upon inactivation of *Smarca4* compared with *Smarcb1* were highly correlated, indicating that *Smarcb1* loss or *Smarca4* loss affect a similar set of enhancers (Supplementary Fig. 8a). However, a small number of loci are reproducibly affected by only *Smarcb1* loss or *Smarca4* loss, suggesting that the effect of losing one subunit of the complex is not identical to the loss of another subunit. (Supplementary Fig. 8b).

Given that the effects of SWI/SNF subunit loss appeared to be localized specifically at enhancers, we next evaluated whether deletion of SWI/SNF subunits also affected H3K4me1, a histone modification associated with both active and poised enhancers. We observed that H3K4me1 levels at enhancers also showed a decrease at enhancers, but to a lesser extent than the decrease in H3K27ac levels (Fig. 1c,d), This finding is consistent with the apparent lack of global change in H3K4me1 upon the loss of SWI/SNF subunits (Supplementary Fig 1).

To investigate whether SWI/SNF is acting directly at active enhancers, we performed ChIP-Seq for the core SWI/SNF subunits SMARCC1 and SMARCA4 in wild-type and *Smarcb1*-deficient cells. Because the genome-wide binding profiles of these two subunits were very similar (Supplementary Fig. 3), we considered the average of the two experiments to be representative of SWI/SNF binding for each condition. Interestingly, we found that SWI/SNF was bound at the vast majority of enhancers in wild-type cells—over 95% of enhancers showed enrichment (IP > input; Fig. 2a and Supplementary Fig. 9a). Deletion of the *Smarcb1* subunit led to a widespread reduction in SWI/SNF binding (Fig. 2b and Supplementary Fig. 9b). The reduction of SWI/SNF binding was notably stronger at enhancers relative to promoters, consistent with the changes observed in the H3K27ac mark at these loci. Furthermore, the enhancers where SWI/SNF binding was strongest in wild-type MEFs and the ones that showed the greatest loss of SWI/SNF binding upon *Smarcb1* deletion also showed the strongest loss of H3K27ac upon

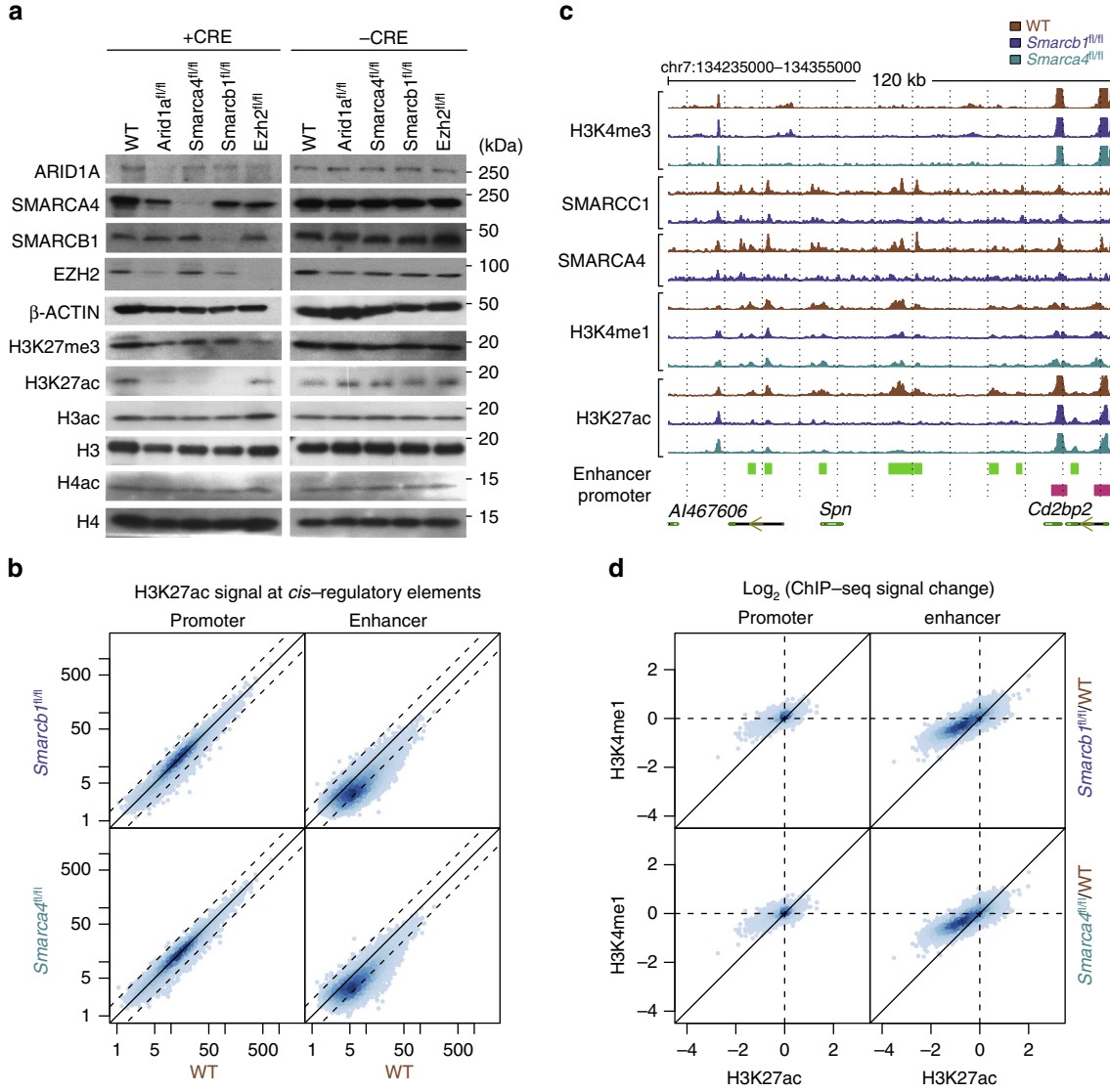

**Figure 1 | Deletion of SWI/SNF subunits results in H3K27ac loss and enhancer deactivation.** (**a**) Western blots of selected factors, histones and histone modifications in control MEFs (-CRE) and MEFs with CRE-inactivated SWI/SNF (*Arid1a*, *Smarca4*, *Smarcb1*) or Polycomb (*Ezh2*) subunits. (**b**). H3K27ac signal at promoters and enhancers in *Smarcb1*-deficient or *Smarca4*-deficient versus wild-type MEFs. (**c**). Representative screenshot showing localization of subunits SMARCC1 and SMARCA4 and histone marks H3K4me3, H3K4me1 and H3K27Ac in wild-type, *Smarcb1*-deficient and *Smarca4*-deficient MEFs showing lost enhancers. (**d**) Log-fold changes of histone modifications H3K4me1 versus H3K27ac at promoters and enhancers in *Smarcb1*-deficient or *Smarca4*-deficient relative to wild-type MEFs.

*Smarcb1* deletion (Supplementary Fig. 10 and Fig. 2c). Taken together, the targeting of SWI/SNF to enhancers and the association between loss of SWI/SNF binding and the loss of H3K27ac suggests a direct role for the SWI/SNF complex in regulating the enhancer chromatin landscape in MEFs.

It is interesting to note that although the majority of enhancers and promoters are bound by SWI/SNF, their sensitivity to deletion of *Smarcb1* was variable both in terms of SWI/SNF binding and the corresponding change in H3K27ac levels (Figs 1b and 2c,d), suggesting that enhancer-specific occupancy of TFs or other co-factors may contribute to the degree with which SWI/SNF is required at different *cis*-regulatory elements. Enhancers are often bound by BRD4 and it was shown recently that super-enhancers, defined as clusters of active enhancers (for example, a high total H3K27ac signal), are more sensitive to BRD4 inhibition than typical enhancers[28] We sought to determine if super-enhancers might also have increased

sensitivity to *Smarcb1* deletion. Surprisingly, we found that the opposite was true: super-enhancers were more refractory to *Smarcb1* deletion, both in terms of SWI/SNF binding and H3K27ac levels (Fig. 2e). Similarly, we identified that enhancers inside and outside transcribed regions (defined by RNA Pol II binding) were also differentially sensitive to *Smarcb1* deletion (Fig. 2e). Like super-enhancer constituents, enhancers in transcribed regions were more likely to retain a residual SWI/SNF complex containing SMARCA4 and SMARCC1 after *Smarcb1* deletion relative to wild-type levels (Supplementary Fig. 11a). Furthermore, those regions were substantially more likely to retain H3K27ac, even after controlling for the smaller change in SWI/SNF binding (Supplementary Fig. 11b). Overall, these findings suggest that SWI/SNF complex is a major regulator of typical distal enhancer activity, and reveal that the activation of some enhancers are less dependent on SWI/SNF, such as those that are within super-enhancers or within transcribed regions.

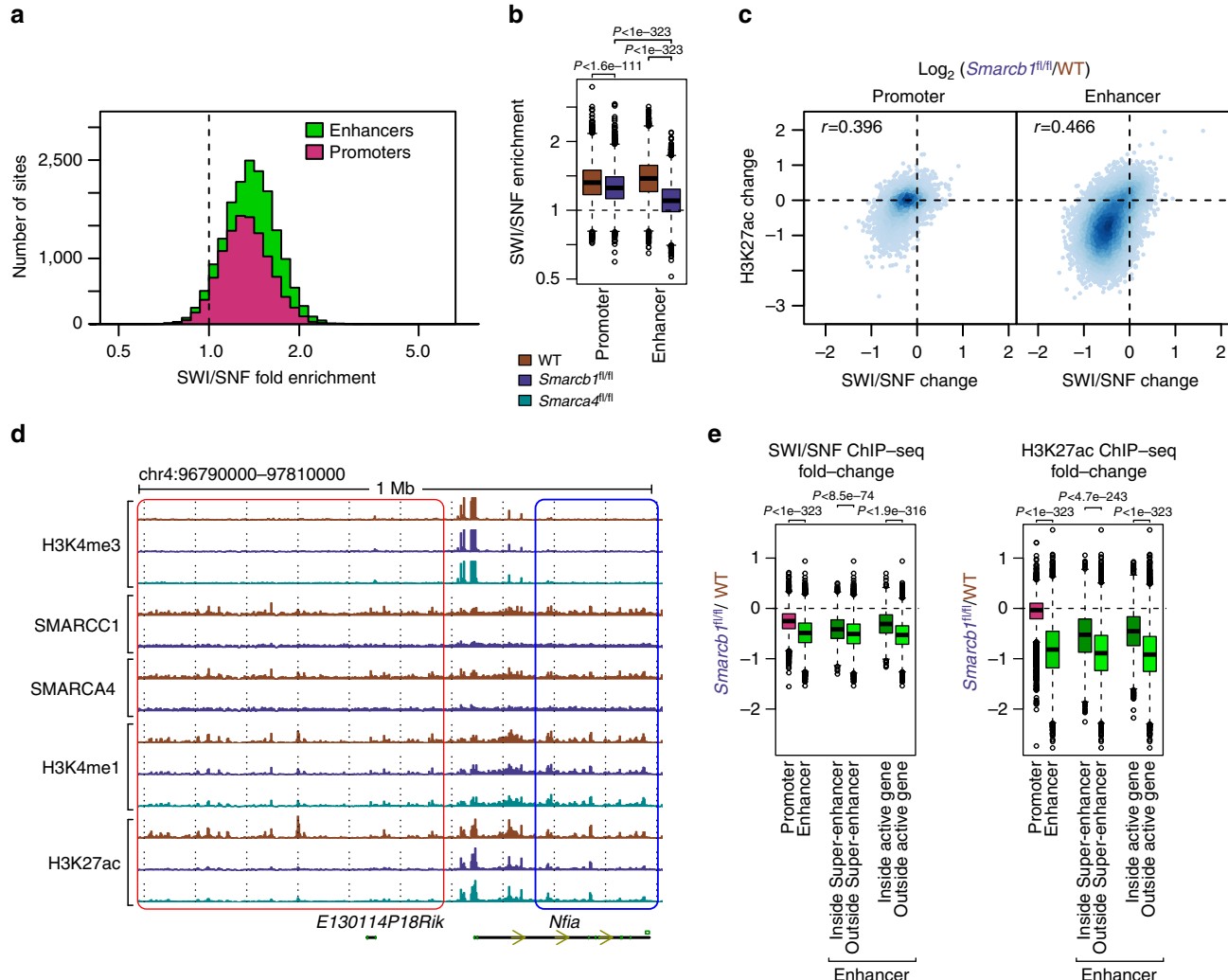

**Figure 2 | SWI/SNF binds most *cis*-regulatory elements in wild-type but typical enhancers are most sensitive to loss of SWI/SNF subunits.**
(**a**) Histogram of average SMARCA4 and SMARCC1 (SWI/SNF) ChIP-seq enrichment over input at promoters and enhancers in wild-type MEFs.
(**b**) Boxplots for SWI/SNF ChIP-seq enrichment over input at promoters and enhancers in wild-type and *Smarcb1*-deficient MEFs. The boxes indicate first,
second and third quartiles, and whiskers show 1.5 × interquartile range below and above the first and third quartile, respectively. Two sided *t*-test *P* values
are shown. (**c**). Fold changes of H3K27ac versus SWI/SNF ChIP-seq signal at promoters and enhancers in *Smarcb1*-deficient relative to wild-type MEFs.
(**d**) Representative screenshot depicting clusters of enhancers with different sensitivities to *Smarcb1* or *Smarca4* loss (red highlights high sensitivity, blue
highlights low sensitivity). (**e**). Boxplots for SWI/SNF and H3K27ac signal fold changes upon *Smarcb1* loss at promoters, enhancers, and enhancers
classified based on super-enhancers (SE) and transcription (PolII enrichment). The boxes indicate first, second and third quartiles, and whiskers show
1.5 × interquartile range below and above the first and third quartile respectively. Two sided *t*-test *P* values are shown.

**SWI/SNF is required for p300 activity at enhancers**. Although
the loss-of-function MEF system enables the evaluation of
SWI/SNF function in a primary cell system that lacks other
mutations, the relevance of these findings to cancer needs to be
separately evaluated. We therefore performed a reciprocal gain-
of-function study in SMARCB1-deficient human RT cell lines via
SMARCB1 re-expression (Fig. 3a, SMARCB1). Consistent with
the dependence of H3K27ac upon *Smarcb1* deletion in MEFs,
re-expression of SMARCB1 in the cancer cell lines resulted
in increased global levels of H3K27ac (Fig. 3a, H3K27ac). Also
consistent with the results obtained in the MEF system, a subtle
increase in H3K4me1 was observed, with no apparent changes in
the total levels of histone H3 or of total H3 acetylation (Fig. 3a).
In addition to the changes in histone modification levels, we
found that the levels of the H3K27 acetyltransferase p300 and
the enhancer-associated factors BRD4 and Mediator were also
increased in the chromatin fractions following induction of
SMARCB1 expression (Fig. 3a).

Although several subunits of the SWI/SNF complex contain
acetyl-lysine binding bromodomains, the SWI/SNF complex does
not contain any known acetyltransferase domains and is thus
unlikely to directly catalyse acetylation of H3K27. Consequently,
we evaluated interactions of SWI/SNF with the H3K27 acetyl-
transferase p300. Consistent with previous reports of physical
interaction between p300 and SWI/SNF[29,30], we found that
multiple SWI/SNF subunits co-immunoprecipitated p300
(Fig. 3b). Notably, however, deletion of either of the *Smarcb1*
or *Smarca4* tumour suppressor subunits in MEFs markedly redu-
ced interactions between p300 and the core SWI/SNF subunit
Smarcc1 (Fig. 3b). Inversely, re-expression of SMARCB1 in RT
cell lines resulted in increase of p300 in the pulldown of the
SMARCC1 subunit (Fig. 3c). Taken together, these results suggest
that the SWI/SNF complex recruits the co-activator p300 to
enhancer loci via a direct interaction.

To further evaluate how the interaction between SWI/SNF and
p300 regulates H3K27ac levels, we immunoprecipitated SWI/SNF

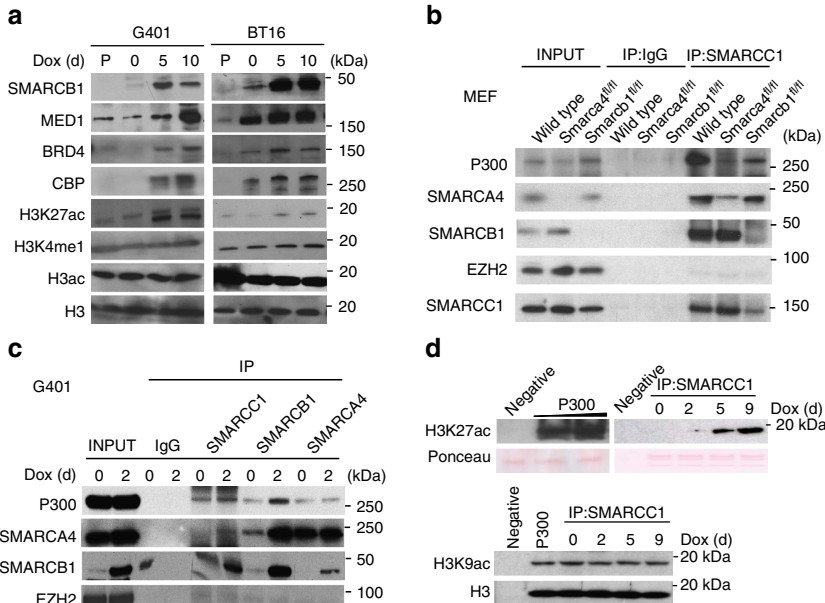

**Figure 3 | The SWI/SNF complex physically interacts with enzymes catalysing histone acetylation at enhancers.** (**a**) Western blots of selected factors, SMARCB1, enhancer-specific binding factors, histone H3, and different modifications in SMARCB1-deficient rhabdoid cell lines, G401 and BT16. Parental lines (P) and 0, 5 or 10 days after Doxycycline (Dox) induced SMARCB1 re-expression. (**b**). Immunoprecipitation (IP) of the SWI/SNF complex subunit SMARCC1 in wild-type and *Smarca4*- or *Smarcb1*-deficient MEFs after CRE-inactivation. Immunoblotted for P300, SMARCA4, SMARCB1, SMARCC1 and EZH2. IgG pulldown is shown as control. (**c**) Immunoprecipitation (IP) of the SWI/SNF complex subunits SMARCC1, SMARCB1 or SMARCA4 from the nuclear extracts of G401 cell line with or without Doxycycline (Dox) induced SMARCB1 re-expression. Immunoblotted for P300, SMARCA4, SMARCB1, and EZH2. IgG is shown as control. (**d**) *In vitro* HAT assay with Acetyl-CoA as donor group and detection of histone acetylation by immunoblot using H3K27ac, H3K9ac, and H3 antibodies. After pulling down the SWI/SNF complex with SMARCC1 antibody, acetylation activity was measured in the presence or absence of SMARCB1 expression. Ponceau staining was shown as loading control for substrate.

complexes using an antibody against the core subunit SMARCC1 and measured histone acetylation activity using recombinant histones as a substrate. With immunoprecipitated SWI/SNF complexes from a RT cell line that lacked SMARCB1, the sample lacked acetyltransferase activity. However, re-expression of SMARCB1 increased H3K27-specific acetyltransferase activity, despite acetylation of H3K9ac being not significantly changed, suggesting the presence of H3K27-specific acetylation activity (Fig. 3d). These results indicate that the SWI/SNF complex is essential for regulation of H3K27ac levels in enhancer regions via protein–protein interactions with the p300 histone acetyltransferase. Inactivation of the SMARCB1 tumour suppressor abrogates this interaction, thereby reducing H3K27ac levels at distal enhancers.

**SWI/SNF mediated enhancer maintenance and cell fate.** In addition to its role as a histone acetyltransferase, p300 has been shown to play key roles in transcription by acting as an adaptor between enhancer-bound TFs and the basal transcription machinery[23,31]. We therefore sought to identify candidate TFs that depend on SWI/SNF to establish enhancer activity. We evaluated enrichment of a set of 4,065 DNA sequence motifs identified by the ENCODE consortium, including a redundant set of TF recognizing elements and their shuffled versions as control[32], in the enhancers that lost H3K27ac signal upon *Smarcb1* loss relative to enhancers that are unaffected (Methods). In enhancers with H3K27ac loss, the most highly overrepresented motifs corresponded to the recognition sequence for the oncogenic heterodimer activator protein-1 (AP-1) TFs (Fig. 4a). The same motifs were also identified in enhancers activated following *SMARCB1* re-expression in SMARCB1-deficient RT cells (Supplementary Fig. 12). The same motifs were also identified in activated enhancers after *SMARCB1* re-expression

in SMARCB1-deficient RT cells (Supplementary Fig. 12). This finding is consistent with a previous study that showed that the BAF60a subunit of SWI/SNF stimulates the DNA binding activity of AP-1 heterodimers[33]. In addition, the ETS motif, which is associated with developmental TFs such as ELF1, PU.1 and FLI1, was also enriched at enhancers sensitive to *Smarcb1* loss in MEFs (Fig. 4a). On the other hand, motifs associated with the insulating factor CTCF were particularly depleted in sensitive enhancers, suggesting that insulator elements, like promoters, may be refractory to *Smarcb1* loss (Fig. 4a and Supplementary Fig. 12).

Finally, to evaluate the transcriptional consequences of the loss of H3K27ac at enhancers after *Smarca4* or *Smarcb1* deletion, we performed RNA-seq analysis in wild-type MEFs and in MEFs lacking *Smarca4* or *Smarcb1*. We observed a significant correlation between the loss of H3K27ac at enhancers and reduced expression of the nearest gene (Fig. 4b). We performed gene ontology (GO) enrichment analysis on genes downregulated following *Smarcb1* or *Smarca4* deletion relative to all expressed genes or relative to genes that are proximal to highly inactivated enhancers. In every analysis, GO terms associated with development and differentiation were among the most significantly enriched (Fig. 4c), consistent with previous findings that the loss of SWI/SNF-dependent enhancer function leads to impaired development and differentiation[9,14,17]. Overall, the requirement of SWI/SNF at typical enhancers and the consequent regulation of gene expression suggests that the loss of differentiation potential is a common mechanism by which SWI/SNF subunit loss leads to tumorigenesis in specific contexts.

## Methods
**Mice.** All mice were housed and cared for according to Animal Care and Use Committee guidelines of the Dana-Farber Cancer Institute (DFCI),

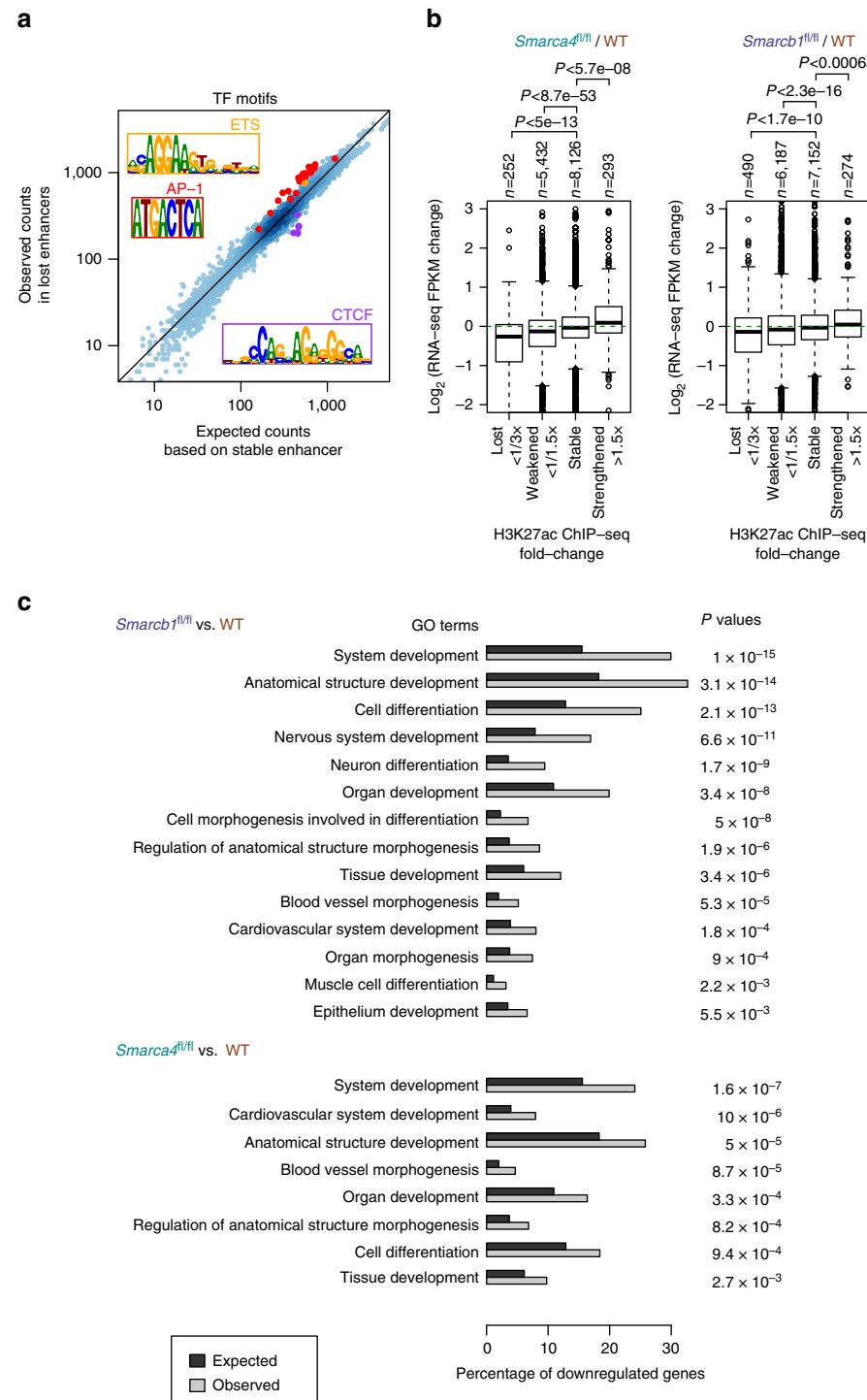

**Figure 4 | SWI/SNF subunit deletion results in downregulation of specific developmental targets.** (**a**). Number of TF motif matches within lost enhancers relative to an expectation based on numbers in stable enhancers. The strongest enrichment/depletion is seen for motifs similar to AP-1, ETS and CTCF motifs, which are highlighted. (**b**). Correlation of nearby gene expression (5–100 kb) changes with H3K27ac signal changes at enhancers upon *Smarcb1* or *Smarca4* deletion. The boxes indicate first, second, and third quartiles, and whiskers show $1.5 \times$ interquartile range below and above the first and third quartile, respectively. Two sided *t*-test *P* values are shown. (**c**) Selected GO terms enriched in downregulated genes. Hypergeometric test *P* values are shown after Benjamini–Hochberg correction for multiple-hypothesis testing.

animal welfare insurance number A3023-01, using DFCI protocol 12-017. The Smarcb1[fl/fl] mice was grown previously in our laboratory[34]; the Smarca4[fl/fl] mice were obtained from Terry Magnuson[35]; the Arid1a[fl/fl] mice were obtained from Zhong Wang[36]; and the Ezh2[fl/fl] mice were obtained from Stuart Orkin[37].

To generate mice that are heterozygous for a *loxP*-flanked allele and hemizygous/heterozygous for the *cre* transgene, a homozygous *loxP*-flanked mouse of the target subunit was mated to a cre transgenic mouse. This yields ∼50% of the offspring as heterozygous for the *loxP* allele and hemizygous/heterozygous for the *cre* transgene. Then, these mice were mated back to the homozygous *loxP*-flanked mice. Approximately 25% of the progeny from this mating are homozygous for the *loxP*-flanked allele and hemizygous/heterozygous for the *cre* transgene. We checked the genotype for each target subunit gene as well as Cre by PCR.

**Cell culture and differentiation.** Primary MEFs were prepared from E13.5 embryos of B6/129 strain wild-type mice or mice that were homozygous for Smarcb1[fl/fl] or Smarca4[fl/fl] or Arid1a[fl/fl] or Ezh2[fl/fl] and hemizygous/ heterozygous for the *cre* transgene. The cells were transduced with pBabe-Cre-puro retrovirus and selected in puromycin $(4 \, \mu g \, ml^{-1})$ for 72 h post infection. G401 cell line was purchased from American Type Culture Collection (ATCC) and BT16 cell line was maintained in the lab. Rhabdoid cell lines were culture in DMEM with 10% FBS at 37 °C with 5% $CO_2$. SMARCB1 doxycycline inducible cell lines were first transduced with lentiviral pInducer21-SMARCB1 plasmid. After 72 h post to transduction, cells were sorted by GFP expression and maintained in Tet-System Approved FBS (Clonetech). SMARCB1 was re-expressed with doxycycline $(1 \, \mu g \, ml^{-1})$ for indicated time.

**Chromatin fractionation.** 72 h post transduction, MEFs and G401 cells were harvested and lysed with buffer A (20 mM HEPES, pH 7.4, 10 mM KCl, 1.5 mM $MgCl_2$, 0.34 M sucrose, 10% glycerol, 1 mM dithiothritol and protease inhibitor cocktail) containing 0.2% Triton X-100. Nuclei were pelleted by centrifugation at 1,000g, resuspended in buffer A without Triton X-100. Next, nuclei were collected and incubated in Buffer B (3 mM EDTA, 0.2 mM EGTA, 1 mM DTT, and protease inhibitor cocktail) for 30 min and centrifuged at 2,000g to remove nuclear debris. After washing once with Buffer B, the samples were resuspended in SDS sample buffer and chromatin associated proteins were analysed by western Blot analysis.

**RNA-seq.** RNAs from two independent biological replicates of primary MEFs (WT, Smarcb1 − / −, and Smarca4 − / −) were extracted with Trizol reagent (Invitrogen) and and further purified using RNeasy MinElute Cleanup Kit (Qiagen). Two microgram of total RNA was used to make the RNA-Seq library using TruSeq Total RNA Sample Prep Kit (Illumina) and sequenced with the Hi-Seq Illumina genome analyser.

**ChIP-seq.** ChIP assays for histone modifications and PolII were performed from $10^7$ MEFs per experiment. Briefly, cells were crosslinked with 1% formaldehyde for 10 min at room temperature and the reaction was quenched by glycine at a final concentration of 0.125 M. Chromatin was digested with MNase to an average size of 1–2 kb. A total of 5 ug of antibody against H3K27ac (Cell Signaling Technology, D5E4, 8173), H3K4me1 (Abcam, ab8895), H3K4me3 (Millipore, 07-473) or PolII (Santa Cruz, N-20, sc-899X, lot#C0813) was added to the digested chromatin (1,000 ul) and incubated overnight at 4 °C.

For the ChIP-seq of SWI/SNF complex subunits, the protocol was slightly modified. In particular, dual crosslinking and sonication were utilized. Cells were first crosslinked in 2 mM disuccinimidyl glutarate (DSG; Life Technologies: Cat. #20593) for 30 min then in 1% formaldehyde for 10 min, quenched with glycine for 5 min. Cells were washed with PBS three times then used to generate nuclear extract. Chromatin was fragmented using sonication based the adaptive focused acoustics technology developed by Covaris. Solubilized chromatin was immunoprecipitated with antibodies against SMARCA4/BRG1 (Abcam: ab110641, 10 µl), SMARCC1/BAF155 (Santa Cruz: sc9746; 10 µg).

For both, protein G Dynal magnetic beads were added to the ChIP reactions and incubated for 4–6 h at 4 °C. Magnetic beads were washed and chromatin was eluted. After crosslinking reversal, RNase A, and proteinase K treatment, ChIP DNA was extracted with the Min-Elute PCR purification kit (Qiagen). ChIP DNA was quantified with Quant-it PicoGreen dsDNA Assay Kit (Life Technologies). 10 ng of ChIP DNA per sample was used to prepare sequencing libraries, and ChIP DNA and input controls were sequenced with the Hi-Seq Illumina genome analyser.

**Immunoprecipitation.** Cre-retrovirus induced MEFs and doxycycline treated rhabdoid cells were washed twice with 1 × PBS and whole cell lysates were prepared using lysis buffer (100 ul; 20 mM HEPES, pH 7.8, 150 mM NaCl, 10 mM EDTA, 2 mM EGTA and 2 mM dithiothreitol, and 0.1% Nonidet P-40). The cell lysates were mixed with Dynabeads conjugated with anti-SMARCC1 (Santa Cruz; A301-021A, 1 ug), anti-SMARCB1 (Bethyl Laboratories; A301-087A, 1 ug), anti-SMARCA4 (Santa Cruz; A300-813A, 1 ug) antibodies and rotated at 4 °C overnight before removal of the supernatant. The resulting samples were suspended SDS gel loading buffer and analysed by western blot analysis using anti-SMARCC1 (Santa Cruz; A301-021A, 1:1,000), anti-SMARCB1 (Bethyl Laboratories; A301-087A, 1:4,000), anti-SMARCA4 (Santa Cruz; A300-813A, 1:500), anti-EZH2 (Cell Signaling Technology; D2C9, 1:1,000), anti-CBP (Cell Signaling technology D6C5, 1:1,000) and anti-P300 (Santa Cruz; sc-585, 1:1,000) antibodies. The raw blot images are shown in Supplementary Fig. 13.

**In vitro histone acetyltransferase assay.** Unmodified full-length recombinant histone H3 was mixed with Acetyl-CoA (Active Motif) in 1 × Assay Buffer in a final volume of 50 uL and incubated at 30 °C for 1 h. Recombinant p300 catalytic domain (Active Motif) was used as a positive control.

**ChIP-Seq sequence alignment and filtering.** The sequenced reads were aligned to the mm9 genome assembly using Bowtie 0.12.6 (ref. 38), allowing up to 10 matches ('-m 10 --best' options)[38]. Regions which had very high signal in the input samples were considered to be spurious and blacklisted: for each input sample, high signal regions were defined as the set of 1 kb sized bins where the signal was more than the mean + 3 s.d. for all the 1 kb sized bins in the genome. The size of these spurious regions varied between 203 and 262 kb for each sample, with a union size of 280 kb.

Reads on the 19 assembled autosomes excluding the 280 kb blacklist region were kept for downstream analysis. The typical fragment size for the different samples ranged between 140 and 180 bp. Each read was considered to represent a signal at half typical fragment size from the 5′-end. Library complexity was calculated for each sample as the number of unique bp positions mapped on each strand, divided by the total number of mapped reads. For batches of experiments where the typical library complexity was below 90% (H3K27ac replicate I and H3K4me1 replicate I, SMARCA4 and SMARCB1), only one read mapping to each position was kept.

**Assessment of ChIP-seq data quality and reproducibility.** To validate the observations of H3K27ac loss at TSS-distal sites upon *Smarcb1* or *Smarca4* loss, we repeated the H3K27ac ChIP-seq experiments on indepedently derived MEFs. In the main text and figures, we show the average of results from the two replicate sets. In this section, we demonstrate the reproducibility of the observations. Genome-wide correlations for ChIP-experiments are shown in Supplementary Fig. 3. The highest genome-wide correlations are seen between replicate experiments: replicates of H3K27ac, replicates of H3K4me1, followed by two subunits of the SWI/SNF complex SMARCA4 and SMARCC1.

To systematically assess the reproducibility of experiments, we follow guidelines determined by the ENCODE consortium[39]. For histone marks with local binding patterns such as acetylation marks or H3K4me2/3, ENCODE uses the R package SPP[40] both to call point binding sites (*find.binding.positions* function) and regions of enrichment (RoE) with a search window of 400 bp (*get.broad.enrichment.clusters* function). For TFs, only the former calls are made. For both histone marks and TFs, reproducibility is evaluated using the irreproducible discovery rate (IDR) measure on point binding calls from replicate experiments[39,41].

In this work, we have utilized the RoE calls to identify *cis*-regulatory elements, since histone marks are not found at fixed single bp resolution binding positions like TFs. However, to obtain IDR values, we have called point binding sites for H3K27ac samples (*find.binding.positions* function) with a loose threshold of false discovery rate < 0.05. Peaks from two replicates were considered to be the same peaks if they were within 90 bp of each other. Peak signal values, $y$, from the two replicates were used to determine IDR using the *est.IDR* function from R package IDR. If a peak in one sample did not match any peaks in the replicate sample, a signal signal value of $y = 0$ was assigned in the replicate. We have also performed an IDR comparison between two pseudoreplicates generated by randomly partitioning the reads from the two replicates. ENCODE guidelines require that the number of peaks with IDR < 0.01 for true replicates ($N_t$) should be more than 50% of the number of peaks for pseudoreplicates ($N_p$)[39]. Our data passed that threshold for all three conditions (Supplementary Table 1). Reproducbility of H3K27ac ChIP-seq experiments was also assessed for the signal at *cis*-regulatory elements, defined below. As expected, we observed that the replicates show a high degree of correlation in terms of the signal at promoters and enhancers (Supplementary Fig. 4).

**Identification of enhancer and promoters.** ChIP-seq RoE for the H3K27ac and H3K4me3 samples were identified using the get.broad.enrichment.clusters function in the SPP[40] package in R with options window.size = 500 and z.thr = 4, relative to matching input samples for each IP experiment. Active TSSs were defined as all TSSs in Ensembl release NCBIM37.67 that overlapped an H3K4me3 RoE in any of the three conditions. Only H3K27ac RoEs which were called in at least two of the six samples (two replicates of three conditions each) were retained for downstream analysis to remove any non-reproducible calls. The union of RoEs for each mark across different samples was used to call enhancers and promoters.

H3K27ac RoEs overlapping both an active TSS and an H3K4me3 RoE were called as promoters. Those more than 1 kb away from an H3K4me3 RoE and more than 2 kb away from an active TSS were called as enhancers. Others were ambiguous and excluded from studies specific to enhancers or promoters.

To validate, we compare our calls to enhancer calls by Mouse ENCODE project which is based on a random forest approach on multiple histone mark data[27]. We observe that a large fraction of our enhancer calls agree with the MEF enhancer from Mouse ENCODE (Supplementary Fig. 2, 15752/21772, 72%), even though different data and very different approaches are used for the calls. Note that, the overlap of our enhancer calls with calls in cell types other than MEFs is much smaller ( <32%).

**Signal enrichment and fold-change at enhancer and promoters.** ChIP-seq signal at each region (enhancer or promoter) was quantified as 'number of reads in region per million mapped reads + pseudocount of 1'. ChIP-seq signal enrichment was then defined as signal in IP sample divided by signal in input signal. ChIP-seq

signal fold-change is defined as IP signal in a given sample divided by IP signal in a control sample. Input signal was not taken into account in the calculation of differences between conditions, because little systematic difference was observed between input samples in different conditions; and the statistical variability gets large if ratios of ratios are calculated.

**Immunoprecipation efficiency correction for ChIP-seq.** While we saw a high degree of correlation between H3K27ac signal as *cis*-regulatory elements between replicates in Supplementary Fig. 4, where a typical normalization based on number of aligned reads in the genome is applied; we noted, that the signal values for replicates do not line up at the diagonal $y = x$, but rather appear to be off by a constant factor. We expect two ChIP-seq libraries that are normalized in this fashion to be comparable if two assumptions hold:

(I) The factor that is being pulled down has a similar level of binding to DNA for the two samples.
(II) The IP efficiency was similar in the two experiments, that is, the libraries contain similar ratio of pulled down to background fragments.

In particular, for replicate experiments of the same condition, the first assumption is known to hold and the validity of the second assumption can be directly assessed. The constant offset we observe for replicate experiments is consistent with variability in the efficacy of IP pulldown between different ChIP-seq experiments.

The need for a more rigorous approach to ChIP-seq library normalization has recently been recognized by multiple groups[42–44]. In this work, we have sought to identify a method to normalize signal values across samples with a data-driven approach with minimal assumptions, while accounting for the variability in pulldown efficiency. When we compare signal values across different conditions, we observe that a large fraction of promoters are observed to lie parallel to the $y = x$ line with a different offset (Supplementary Figs 5 and 6). A number of lines of evidence suggests that the real levels of H3K27ac are unchanged at a large fraction of promoters upon *Smarcb1* or *Smarca4* deletion:

• A large fraction of promoters show a very narrow distribution of fold-difference values (Supplementary Fig. 5). It is unlikely that all promoters are affected at the same constant fold-change level with very little variability.
• For each mutant, one of the replicates shows a distribution of fold-differences around 1 (*Smarcb1*/wild-type, repeat I and *Smarca4*/wild-type, repeat II).
• The two comparisons for which a constant shift is seen (*Smarcb1*/wild-type, repeat II and *Smarca4*/wild-type, repeat I) show higher H3K27ac signal at promoters upon SWI/SNF subunit inactivation. This is the opposite trend to what is seen for global H3K27ac levels in western blots (See Fig. 1a).
• Gene expression changes are consistent with a picture where the H3K27ac signal at the bulk of promoters are unchanged. Genes near enhancers or promoters with little change after a promoter-based normalization show no expression change on average (See Fig. 4b for enhancers; promoters not shown).

On the basis of these observations, we applied an additional multiplicative factor to each H3K27ac sample after library size normalization to set the mode of the log-fold-change distribution at promoters to zero while comparing different samples. These factors were: WTI = 1.28, *Smarca4*I = 0.85, *Smarcb1*I = 1.31, WTII = 1.14, *Smarca4*II = 1.14, *Smarcb1*II = 0.66. Supplementary Fig. 7 shows comparison of different samples after applying this multiplicative correction.

We refrained from applying a similar normalization for other ChIP-seq sample sets, since we could not confidently determine a set of regions where they are unaffected upon *Smarcb1* or *Smarca4* deletion. For these data, we follow library size normalization as our best estimate on how samples should be compared.

**Gene ontology analysis for enhancers.** Each enhancer was associated to the closest active TSS within 100 kb. Enhancers sensitive to *Smarcb1* or *Smarca4* knockout were selected based on two criteria: (i) the signal change mutant/WT was below 1 for each replicate set; and (ii) the geometric mean of the signal change for the two replicate comparisons was below ½. $P$ values for gene set enrichment for genes associated to sensitive enhancers were calculated relative to genes associated with any enhancer using hypergeometric test. $q$ values were obtained based on Benjamini–Hochberg procedure.

**ChIP-seq visualization.** Genomic profiles for visualization were generated using Gaussian smoothing with $\sigma = 100$ bp after library size normalization (for example, in Fig. 1c, or output wig files).

**Identification of super-enhancers.** Super-enhancers were called with a slightly modified approach from the original method[20]. H3K27ac RoEs were called as described above. We did not remove TSS-proximal peaks, but stitched all RoEs

within 12.5 kb. For each stitched RoE, IP and input signal were calculated only in portions that did not intersect H3K4me3 RoEs (as defined above). We found this approach to be better at removing false positives from stitched enhancer peaks which encompassed active TSSs. Only H3K27ac RoEs which were called in at least two of the six samples (two replicates of three conditions each) were retained to remove any non-reproducible calls.

**Identification of transcribed regions.** The genome was segmented based on PolIII enrichment in wild-type using the BIC-seq command line tool MBICseq[45] with option 'l 1', and providing PolII and input read counts in 100 bp bins. Segments with average log2-enrichment greater than 0.1 were selected as transcribed regions.

**Transcription factor motif enrichment in human cell lines.** TF motif maps for hg19 for 4,095 motifs (including a redundant set of real TF recognition elements, and shuffled motif control sequences) were downloaded from http://compbio.mit.edu/encode-motifs/[46]. The position weight matrix (p.w.m.) for each motif was calculated based on the actual sequences of the provided motif locations. The number of motif occurrences was counted for each motif inside sensitive enhancers (H3K27ac fold-change > 2) and insensitive enhancers (1/1.5 < H3K27ac fold-change < 1.5). A lowess curve was calculated to model the ratio of counts for each motif (sensitive/insensitive) as a function of the GC content of the motif p.w.m. This curve was used to calculate back the expected number of occurrences for each motif. Fig. 4a and Supplementary Fig. 12 show observed counts versus expected counts for the 4,095 motifs. p.w.m. for two selected motifs ('AP-1_known3_8mer' and 'CTCF_known1_8mer') are displayed on the figure. Motif similarity was assessed based on Pearson correlation values between motif p.w.m., Motifs which are similar to the two selected motifs ($r > 0.85$) are highlighted on the figures.

**Transcription factor motif enrichment in MEFs.** To perform the same analysis for MEFs, we first sought to create a corresponding motif map for the mouse genome build mm9. For each motif, we utilized the p.w.m. calculated above based on the provided hg19 map. The locations of motif matches were determined using the MEME suite tool mast with default parameters[47]. A further selection on the motif matches was applied to mimic the results obtained for hg19. For each motif, a threshold on the match $P$ value was set such that the number of matches to mm9 was barely as many as the number of matches to the hg19 map. Any matches for the motif below this threshold was discarded.

The experiments on the human samples were performed once, whereas the experiments for the MEF samples were in replicates. The same approach as used for human cell lines was applied for MEFs, taking advantage of the replicate pairs, to compare highly sensitive enhancers (more than 2× reduction in H3K27ac signal in each replicate pair and more than 4x reduction on average) and insensitive enhancers (<2× change in H3K27ac signal in each replicate, and <1.5-fold-change on average).

**RNA-seq sequence alignment and gene level quantification.** The sequenced reads from each sample were aligned to the mouse genome + transcriptome assembly NCBIM37.67 using TopHat v2.0.8 with default parameters except turning off novel junction search ('-G <gtf> --no-novel-juncs' options)[48]. The transcriptome was self-merged to allow processing with cufflinks v2.1.1 tool cuffdiff, "cuffcompare -s mm9.fa -CG -r NCBIM37.67.gtf NCBIM37.67.gtf"[48]. Different conditions were compared using cuffdiff with default parameters and bias correction ('-G <gtf> -b' options).

**Relating changes at enhancer to gene expression.** Each enhancer was associated to the closest active TSS, as defined above. The ratio of IP signal for mutant divided by WT MEFs was used to categorize enhancers to four groups: more than 3-fold signal loss, between 1.5- and 3-fold signal loss, less than 1.5-fold-change, and more than 1.5-fold signal increase. The change in RNA-seq was quantified as log2 (normalized gene level count value from cuffdiff + pseudocount of 5, in mutant/the same in WT). Only enhancers for which the closest active TSS was 5–100 kb away, and the corresponding gene had cuffdiff status 'OK' were retained.

**Gene ontology enrichment analysis of downregulated genes.** Gene level comparisons output by cufffdiff (gene_exp.diff) were used for GO analysis. For each mutant / WT comparison, only genes with status = OK were retained. Downregulated genes were selected as those reported by cuffdiff as significantly different (false discovery rate < 0.05) and with a greater than 2-fold reduction in gene expression when comparing mutant versus WT. GO databases were downloaded from http://geneontology.org on 29 April 2014. $P$ value for GO term enrichment in downregulated genes versus all (status = OK) genes was calculated using hypergeometric test. Multiple-hypothesis correction was applied using Benjamini–Hochberg procedure. Selected terms are displayed in the figures.

**Data availability.** All sequencing data have been submitted to the GEO database with accession number GSE71509. All other data are available from authors upon reasonable request.

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

## Acknowledgements

We thank members of the Roberts and Park laboratories for assistance and discussion. In particular, we thank Soohyun Lee and Ryan Lee for careful reading of the manuscript. This work was supported by the US National Institutes of Health grants R01CA172152 (C.W.M.R.) and R01CA113794 (C.W.M.R.). K.H.K. was supported by an award from National Cancer Center. X.W. was supported by the Pathway to Independence Award from the US National Institutes of Health (K99CA197640), a postdoctoral fellowship from the Rally Foundation for Childhood Cancer Research and The Truth 365, and a research grant from the St. Baldrick's Foundation. The Avalanna Fund, the Cure AT/RT Now Foundation, the Garrett B. Smith Foundation, Miles for Mary, and ALSAC/St. Jude (C.W.M.R.) provided additional support.

## Author contributions

B.H.A., K.H.K., P.L., X.W., P.J.P. and C.W.M.R. conceived experiments and study design and interpreted the results. K.H.K., P.L., X.W., H.E.M., W.W. and J.R.H. performed all the experiments. B.H.A. performed computational and statistical analyses of the data. B.H.A., K.H.K., P.L., X.W. and C.W.M.R. wrote the manuscript with input from all co-authors.
