## [Peer Review File · Nature Communications]

REVIEWERS' COMMENTS:

Reviewer #1 (Remarks to the Author):

In this work, the authors studied the roles of SWI/SNF complex in chromatin state and transcription using conditional KO of Smarca4 (BRG1), Smarcb1 (SNF5/BAF47/INI1), and Arid1a MEFs. Much of analysis was focused on Smarca4 and Smarcb1 KO. In separate manuscripts they discussed Arid1a cKO cells and phenotypes and Smarcb1 KO in cancer. In the current work, they found that the bulk H3K27Ac level was reduced in the KO cells. They did ChIP-seq and found that it was mainly the distal enhancers rather than the promoters whose H3K27Ac levels were reduced. They also found some modest effects of SWI/SNF KO on H3K4Me1 levels at enhancers, but not as pronounced as the effects on H3K27Ac. This was also supported by the finding that there was little effect on the bulk H3K4Me1 level when analyzed by Western blot. Finally, by RNA-seq analysis, the authors showed general reduction of transcription from genes near the K27ac sites that are lost in the KO cells.

Interestingly, super enhancers were not as affected by the SWI/SNF KO as typical enhancers. This is in contrast to the study of BRD4 inhibition, in which super enhancers were found to be more sensitive than typical enhancers.

In addition to the MEFs with SWI/SNF KO, they also studied SMARCB1-deficient rhabdoid tumor cell lines. SMARCB1 re-expression in these cells resulted in increased level of bulk H3K27Ac. This is consistent with the MEF study. They speculate that the effect of SWI/SNF KO on H3K27Ac is due to the interaction of SWI/SNF with p300. To support this model, they showed that p300 (and HAT activity) co-immunoprecipitated with SWI/SNF.

Overall, the authors have provided strong experimental data to show the connection between SWI/SNF complex and H3K27Ac at enhancers. It is interesting that typical enhancers are more affected than promoters in terms of K27ac than super enhancers. Some concerns have to do with the statements.

1. Page 5, line 109. The authors claimed that "the loss of H3K27ac suggests a direct role of the SWI/SNF complex in regulating enhancer activity in MEFs". The problem is two fold. First, is K27ac level a reliable indicator of enhancer activity? I think the jury is still out there. Second, the authors have yet to show even a single case about the distal K27ac sites functioning as enhancers in the traditional sense. It is better to tone down the statement and not use the term "enhancer activity" when the data is at best indirect.

2. Page 8, line 175. The authors stated that "4,065 DNA sequence motifs from the Encyclopedia of DNA Elements". This number is surprisingly high - how many of them are redundant motifs? The authors need to collapse all the redundant motifs and redo the analysis.

3. In the RNA-seq analysis (Figure 4b), are the K27ac sites analyzed filtered for TSS?

Reviewer #2 (Remarks to the Author):

I have reviewed the submission by Alver et al as a revision of the manuscript originally submitted to Nature Genetics. My technical concerns from the first round of review have been addressed. The Supplemental Text on normalization was greatly appreciated and should enable readers who are concerned to better understand the analyses that were performed. The same is true for the additional supplemental figures. I'm still ambivalent about the overall significance of the findings, but the work is done well and the other reviewer is supportive. Maybe I'm just getting old...

Anyway, I can recommend publication once the following minor details are addressed.

- I disagree with the conclusion based on Fig. 3c that states that re-expression of SMARCB1 in RT cell lines resulted in "increased interaction between SWI/SNF and p300". Pulldown with SMARCA4 did not increase the amount of p300 on day 2 compared to day 0, and if there is an increase in the amount of p300 in the day 2 pulldown using SMARCC1, it is very modest. The only convincing result is from the SMARCB1 pulldown. Is there an explanation for these differences that could be shared in the text?
- Please explicitly indicate the antibodies used for ChIP. The "immunoprecipitation" section of the methods lists antibodies for IP and the vendor. Please list the catalog number as well. Were the same antibodies used for ChIP?
- the sentence on lines 172-173 is repeated on lines 173-174.
- typo in Figure 1a in the column of tested proteins – presumably SMARCA4, not SMARC1A.
- typo in Supp. Fig. 1 in the column of tested proteins – SMARCB1, not SMARBC1.

Reviewers' comments and our responses.

Reviewer #1 (Remarks to the Author):

In this work, the authors studied the roles of SWI/SNF complex in chromatin state and transcription using conditional KO of Smarca4 (BRG1), Smarcb1 (SNF5/BAF47/INI1), and Arid1a MEFs. Much of analysis was focused on Smarca4 and Smarcb1 KO. In separate manuscripts they discussed Arid1a cKO cells and phenotypes and Smarcb1 KO in cancer. In the current work, they found that the bulk H3K27Ac level was reduced in the KO cells. They did ChIP-seq and found that it was mainly the distal enhancers rather than the promoters whose H3K27Ac levels were reduced. They also found some modest effects of SWI/SNF KO on H3K4Me1 levels at enhancers, but not as pronounced as the effects on H3K27Ac. This was also supported by the finding that there was little effect on the bulk H3K4Me1 level when analyzed by Western blot. Finally, by RNA-seq analysis, the authors showed general reduction of transcription from genes near the K27ac sites that are lost in the KO cells.

Interestingly, super enhancers were not as affected by the SWI/SNF KO as typical enhancers. This is in contrast to the study of BRD4 inhibition, in which super enhancers were found to be more sensitive than typical enhancers.

In addition to the MEFs with SWI/SNF KO, they also studied SMARCB1-deficient rhabdoid tumor cell lines. SMARCB1 re-expression in these cells resulted in increased level of bulk H3K27Ac. This is consistent with the MEF study. They speculate that the effect of SWI/SNF KO on H3K27Ac is due to the interaction of SWI/SNF with p300. To support this model, they showed that p300 (and HAT activity) co-immunoprecipitated with SWI/SNF.

Overall, the authors have provided strong experimental data to show the connection between SWI/SNF complex and H3K27Ac at enhancers. It is interesting that typical enhancers are more affected than promoters in terms of K27ac than super enhancers. Some concerns have to do with the statements.

1. Page 5, line 109. The authors claimed that "the loss of H3K27ac suggests a direct role of the SWI/SNF complex in regulating enhancer activity in MEFs". The problem is two fold. First, is K27ac level a reliable indicator of enhancer activity? I think the jury is still out there. Second, the authors have yet to show even a single case about the distal K27ac sites functioning as enhancers in the traditional sense. It is better to tone down the statement and not use the term "enhancer activity" when the data is at best indirect.

In the mentioned section, we find that i. SWI/SNF binds to enhancers, ii. the enhancers with highest SWI/SNF binding lose K27ac most, iii. The enhancers which

most strongly lose SWI/SNF binding lose K27ac most. These observations do suggest a direct link between the loss of SWI/SNF subunits and the loss of K27ac at enhancers. In response to the reviewer's point, we have reworded the sentence to avoid the extrapolation to "enhancer activity" and instead state a role in "regulating the enhancer chromatin landscape". We have also reworded the title of the paper along the same lines. We should note that in the following sections of the paper, we do show a highly significant correlation between loss of K27ac at enhancers and nearby gene expression changes.

2. Page 8, line 175. The authors stated that "4,065 DNA sequence motifs from the Encyclopedia of DNA Elements". This number is surprisingly high - how many of them are redundant motifs? The authors need to collapse all the redundant motifs and redo the analysis.

The reviewer is right that this motif set is highly redundant. In fact, the analysis we perform relies on this redundancy to better identify sequence biases, and including fake motifs in the analysis provides an independent confirmation of our false discovery rate estimates. In the end, we report sets of redundant motifs together, e.g. motifs similar to the AP1 recognition element. The details are described in the methods section. We have also reworded the sentence in the main text to clarify that the approach was a conscious choice.

3. In the RNA-seq analysis (Figure 4b), are the K27ac sites analyzed filtered for TSS?

Yes. This filtering was already described in the methods. We have now added "5-100kb" in the legend as well.

Reviewer #2 (Remarks to the Author):

I have reviewed the submission by Alver et al as a revision of the manuscript originally submitted to Nature Genetics. My technical concerns from the first round of review have been addressed. The Supplemental Text on normalization was greatly appreciated and should enable readers who are concerned to better understand the analyses that were performed. The same is true for the additional supplemental figures. I'm still ambivalent about the overall significance of the findings, but the work is done well and the other reviewer is supportive. Maybe I'm just getting old...

Anyway, I can recommend publication once the following minor details are addressed.

- I disagree with the conclusion based on Fig. 3c that states that re-expression of SMARCB1 in RT cell lines resulted in "increased interaction between SWI/SNF and

p300". Pulldown with SMARCA4 did not increase the amount of p300 on day 2 compared to day 0, and if there is an increase in the amount of p300 in the day 2 pulldown using SMARCC1, it is very modest. The only convincing result is from the SMARCB1 pulldown. Is there an explanation for these differences that could be shared in the text?

As the reviewer commented, we have also noticed that there are some differences in the integrity of the pulldown when antibodies recognizing different subunits were used. Noting the the lysate for IP was not crosslinked, there are a number of possible scenarios to explain this difference, having to do with the composition and abundance of the SWI/SNF complex with or without the SMARCB1 subunit, or the nature of the interaction between p300 and SWI/SNF and the antibody recognition sites. We cannot resolve the different scenarios with our current data. Therefore, we have decided not to speculate. We have changed the text to more directly state our observations: " Inversely, re-expression of SMARCB1 in RT cell lines resulted in increase of p300 in the pulldown of the SMARCC1 subunit (Fig. 3c)."

• Please explicitly indicate the antibodies used for ChIP. The "immunoprecipitation" section of the methods lists antibodies for IP and the vendor. Please list the catalog number as well. Were the same antibodies used for ChIP?

All the antibody information is now included in the methods.

- the sentence on lines 172-173 is repeated on lines 173-174.
- typo in Figure 1a in the column of tested proteins – presumably SMARCA4, not SMARC1A.
- typo in Supp. Fig. 1 in the column of tested proteins – SMARCB1, not SMARBC1.

These typos have been fixed.